



# Measurement report: Evolution and distribution of NH₃ over Mexico City from ground-based and satellite infrared spectroscopic measurements

Beatriz Herrera[1,2,3], Alejandro Bezanilla[1], Thomas Blumenstock[4], Enrico Dammers[5], Frank Hase[4], Lieven Clarisse[6], Adolfo Magaldi[7], Claudia Rivera[1], Wolfgang Stremme[1], Kimberly Strong[3], Camille Viatte[8], Martin Van Damme[6,9], and Michel Grutter[1]

[1]Instituto de Ciencias de la Atmósfera y Cambio Climático, Universidad Nacional Autónoma de México, Mexico City, 04510, Mexico
[2]Department of Physical and Environmental Sciences, University of Toronto Scarborough, Toronto, M1C 1A4, Canada
[3]Department of Physics, University of Toronto, Toronto, M5S 1A7, Canada
[4]Karlsruhe Institute of Technology (KIT), Institute of Meteorology and Climate Research (IMK-ASF), Karlsruhe, Germany
[5]Climate, Air and Sustainability (CAS), Netherlands Organisation for Applied Scientific Research (TNO), Utrecht, Netherlands
[6]Spectroscopy, Quantum Chemistry and Atmospheric Remote Sensing (SQUARES), Université libre de Bruxelles (ULB); Brussels B-1050, Belgium.
[7]ENES Juriquilla, Universidad Nacional Autónoma de México, Querétaro, 762630, Mexico
[8]LATMOS/IPSL, Sorbonne Université, UVSQ, CNRS, Paris, France
[9]Royal Belgian Institute for Space Aeronomy (BIRA-IASB), Brussels, Belgium

*Correspondence to*: Beatriz Herrera (beatriz.herrera@mail.utoronto.ca)

**Abstract.** Ammonia (NH₃) is the most abundant alkaline compound in the atmosphere, with consequences for the environment, human health, and radiative forcing. In urban environments, it is known to play a key role in the formation of secondary aerosols through its reactions with nitric and sulphuric acids. However, there are only a few studies about NH₃ in Mexico City. In this work, atmospheric NH₃ was measured over Mexico City between 2012 and 2020 by means of ground-based solar absorption spectroscopy using Fourier transform infrared (FTIR) spectrometers at two sites (urban and remote). Total columns of NH₃ were retrieved from the FTIR spectra and compared with data obtained from the Infrared Atmospheric Sounding Interferometer (IASI) satellite instrument. The diurnal variability of NH₃ differs between the two FTIR stations and is strongly influenced by the urban sources. Most of the NH₃ measured at the urban station is from local sources, while the NH₃ observed at the remote site is most likely transported from the city and surrounding areas. The evolution of the boundary layer and the temperature play a significant role in the recorded seasonal and diurnal patterns of NH₃. Although the vertical columns of NH₃ are much larger at the urban station, the observed annual cycles are similar for both stations, with the largest values in the warm months, such as April and May. The IASI measurements underestimate the FTIR NH₃ total columns by an average of 32.2 ± 27.5 % but exhibit similar temporal variability. The NH₃ spatial distribution from IASI shows the largest columns in the northeast part of the city. In general, NH₃ total columns over Mexico City exhibited an



average annual increase of $92 \pm 3.9 \times 10^{13}$ molecules/cm² yr (urban) and $8.4 \pm 1.4 \times 10^{13}$ molecules/cm² yr (remote) was

observed in Mexico City at both FTIR stations and a decadal increase of 62 % with IASI data.

# 1 Introduction

Atmospheric ammonia ($NH_3$) is the most abundant basic gas in ambient air (Behera et al., 2013). It predominantly reacts with sulfuric acid ($H_2SO_4$) and nitric acid ($HNO_3$) vapor to neutralize a significant fraction of the atmospheric acidity and form ammonium sulfate and ammonium nitrate salts (Seinfeld and Pandis, 2006). Rich $NH_3$ environments thus promote the

formation of secondary inorganic aerosols, which can account for up to 50% of the mass in the total particular matter (PM) (Behera et al., 2013). $PM_{2.5}$ (PM with an aerodynamic diameter < 2.5 µm) is associated with premature human mortality (Paulot and Jacob, 2014; Giannakis et al., 2019), highlighting the importance of taking action to reduce the health impacts due to air pollution, particularly in densely populated environments.

$NH_3$ has a short lifetime, on the order of hours to a few days (Dammers et al., 2016, 2019; Nair and Yu, 2020; Evangeliou et

al., 2021), and exhibits a strong temporal and spatial variability that ranges over three orders of magnitude near the surface (Shephard et al., 2011). $NH_3$ emissions and deposition strongly depend on environmental conditions. The primary sources of atmospheric $NH_3$ are related to agricultural activities (mainly livestock and fertilizers), as well as natural sources, biomass burning, vehicular emissions, humans and pets (Bouwman et al., 1997; Sutton et al., 2008, 2013). Human $NH_3$ emissions strongly depend on temperature and skin exposure, for example, one adult can emit 0.4 mg of $NH_3$ per hour at 25 ˚C but 1.4

of $NH_3$ per hour at 29 ˚C (Li et al., 2020). In countries with intensive livestock production, $NH_3$ is the main contributor to nitrogen fluxes. Emitted $NH_3$ can be transported by winds and removed from the atmosphere by wet and dry deposition (Neirynck and Ceulemans, 2008; Behera et al., 2013). $NH_3$ deposition also has an important role in the acidification and eutrophication of ecosystems (Krupa, 2003; Sutton et al., 2008), with multiple effects on water, air, soil, climate and biodiversity (Sutton et al., 2013).

$NH_3$ accounts for almost half of all reactive nitrogen released in the atmosphere, with total $NH_3$ emissions doubling from 1860 to 1993 and possibly doubling again by 2050 (Krupa, 2003; Galloway et al., 2004; Clarisse et al., 2009) mainly driven by the increasing use of fertilizers. Recent advances in satellite remote sensing have resulted in a better knowledge of global $NH_3$ concentrations, however, uncertainties in the total $NH_3$ budget, along with the specific emission sources across different spatial scales, remain high mainly due to the lack of observations on land (Clarisse et al., 2009; Behera et al., 2013; Sutton

et al., 2013).

The number and size of the world's cities is increasing, with some of them becoming megacities, hosting more than 10 million inhabitants. The urban population worldwide is expected to continue this increase in the coming years, adding about ten more megacities by 2030 (United Nations, 2018). These massive concentrations of people and their activities present





significant challenges for the global environment, especially in terms of air pollution, climate, and human health. One of the
largest metropolitan areas in the world, and the largest in North America, is the Mexico City Metropolitan Area (MCMA), a
megacity of >21 million inhabitants that presents poor air quality during many days of the year. It is located in a basin
surrounded by mountains and volcanoes, complicating the ventilation of the polluted air (Molina et al., 2020) that is
dominated by the dynamics of the boundary layer (Stremme et al., 2013; Dammers et al., 2016). The Mexico City Emissions
Inventory for 2016 (SEDEMA, 2018), reports that the MCMA hosts almost 6 million vehicles and 2300 regulated industries,
and emits a total of 14895 tonnes of $NH_3$/yr only in the Mexico City's area and 47717 tonnes of $NH_3$/yr in the MCMA,
including part of the Estado de Mexico. According to the inventory, 0.1% of $NH_3$ emissions in Mexico City are coming from
"point sources" such as industry, 6% from "mobile sources" such as vehicles, and 94% from "area sources" including urban
waste (0.12%), agriculture (1.66%), livestock (2.42%) and other (89%); within the other category are: urban and forest fires
and paved and unpaved roads. The inventory also strongly attributes the $NH_3$ sources to a range of population activities and
feces from domesticated animals. Despite the frequent pollution episodes due to PM, the local government has not
implemented policies regulating $NH_3$ emissions.

A few studies have investigated atmospheric $NH_3$ in the Mexico City area. Surface $NH_3$ concentrations between 10 and 40
ppbv were measured using an open-path Fourier transform infrared (FTIR) spectrometer, with the highest mixing ratios
observed in the morning hours during a two-month period (Moya et al., 2004). FTIR-$NH_3$ time series between 2012 and
2015 contributed to a validation study of Infrared Atmospheric Sounding Interferometer (IASI) (Dammers et al., 2016) and
Cross-track Infrared Sounder (CrIS) (Dammers et al., 2017) $NH_3$ satellite products. In terms of $NH_3$ emissions in Mexico
City, Yokelson et al. (2007) reported $NH_3$ emission factors from forest fires in the mountains surrounding Mexico City in
2006 and Christian et al., (2010) reported emission factors from garbage burning and domestic and industrial biofuel use in
central Mexico. A more recent study by Cady-Pereira et al. (2017) investigated the impact of biomass burning events on
pollution over the MCMA using trace gas data, including $NH_3$, from the Tropospheric Emission Spectrometer (TES)
instrument onboard the Aura satellite. That study concluded that biomass burning events can impact pollution levels in
Mexico City, specifically the south part of the MCMA and particularly during the March-April-May period. Recently,
Clarisse et al. (2019) and Viatte et al. (2022) reported $NH_3$ hotspots near Mexico City at Tochtepec (18.84°N, 97.80°W),
Ezequiel Montes (20.68°N, 99.93°W), and Tehuacan (18.45°N, 97.31°W), with all of them classified as agricultural sources.
Finally, Van Damme et al. (2021) reported an increasing trend of $NH_3$ over Mexico of $(2.5\pm1.5)\times10^{13}$ molecules cm$^{-2}$ yr$^{-1}$
using 11 years of IASI satellite data (2008-2018).

In this work, the diurnal and seasonal variability of $NH_3$ over Mexico City is investigated using datasets from two ground-
based FTIR spectrometers, including an extension of the FTIR-$NH_3$ total column time series of the station in Mexico City
used in Dammers et al. (2016), and of the FTIR-$NH_3$ total columns measured at Altzomoni, a remote high-altitude station
close to Mexico City, that are retrieved for the first time. The locations of these two sites are shown in Figure 1. The analysis





is complemented with IASI satellite observations over the region, and back-trajectories that were constructed for anomalous NH$_3$ columns detected at the urban site to assess the influence of local and remote sources.

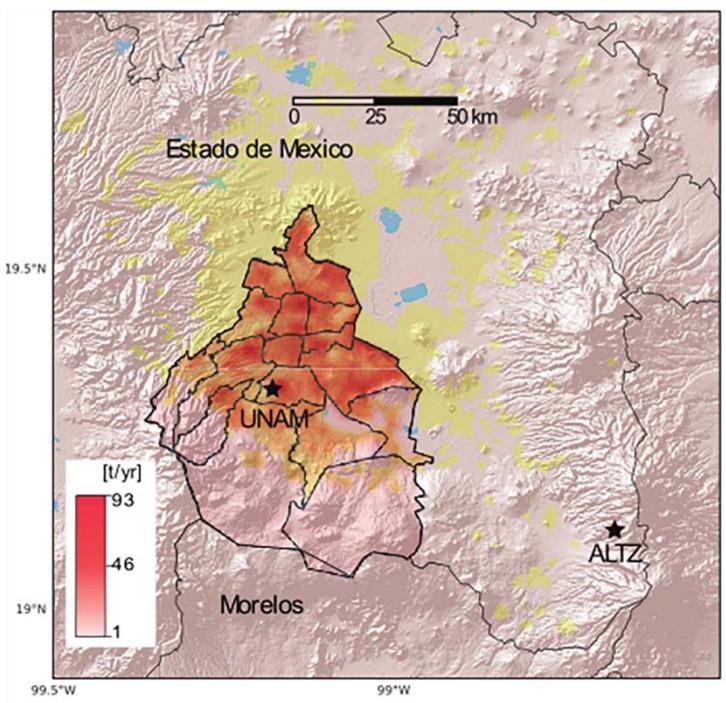

**Figure 1:** Study area in central Mexico. Mexico City is shown in the middle, with the red shading corresponding to NH$_3$ emissions reported in the Mexico City Emissions Inventory 2016 (SEDEMA, 2008) in tons per year. The stars indicate the location of the UNAM and Altzomoni stations and the yellow shading indicates the extension of the MCMA.

## 2 Methodology

### 2.1 Ground-based FTIR stations and retrieval of NH$_3$ total columns

This study utilizes NH$_3$ total columns retrieved from solar absorption spectra measured with ground-based FTIR spectrometers at two sites in and around the MCMA. The urban FTIR station is located at the south of Mexico City within the campus of the Universidad Nacional Autónoma de Mexico on the rooftop of the Instituto de Ciencias de la Atmósfera y Cambio Climático (UNAM, 19.33°N, 99.18°W, 2280 m.a.s.l.). A custom-built solar tracker directs solar radiation to the entrance of a FTIR spectrometer (Bruker Optik GmbH model Vertex 80) that has a maximum unapodized resolution of 0.06 cm$^{-1}$. The instrument is equipped with a KBr beamsplitter and two detectors (HgCdTe and InGaAs). The HgCdTe detector is cooled with liquid nitrogen. For more details about this system, see Bezanilla et al. (2014). Measurements from the remote FTIR site were made at the Altzomoni Atmospheric Observatory (ALTZ, 19.12°N, 98.66°W, 3985 m.a.s.l.), a high-altitude station located 60 km from the UNAM urban site and within the Izta-Popo National Park surrounded by nature. This site is part of the Network for the Detection of Atmospheric Composition Change (NDACC) (De Mazière et al., 2018) contributing





data from a high-resolution FTIR spectrometer since 2012. This instrument (Bruker Optik GmbH model IFS 120/125 HR)
can record solar spectra with a maximum spectral resolution of 0.0035 cm$^{-1}$, and is equipped with KBr and CaF$_2$
beamsplitters and HgCdTe, InSb, and InGaAs detectors. For details of the site and the instrument see Baylon et al. (2017).

The FTIR spectra used for the retrieval of NH$_3$ were collected with the HgCdTe detector at both sites with a spectral
resolution of 0.005 cm$^{-1}$ at Altzomoni, and 0.1 cm$^{-1}$ (prior to 2014) and 0.075 cm$^{-1}$ at UNAM, using an optical band pass
filter to enhance the region between 700 and 1400 cm$^{-1}$ In this study, we extend the UNAM time series used in a previous
2012-2015 comparison between IASI and FTIR (Dammers et al., 2016) with improvements to the retrieval, and we present
for the first time, Altzomoni FTIR-NH$_3$ retrievals for the period between April 2012 and May 2020. The analysis presented
here focuses on the region around the MCMA, also using the IASI satellite product and a back-trajectory evaluation, as
described below.

For both sites, the solar FTIR spectra were analysed using PROFFIT version 9.6 for the retrievals (Hase et al., 2004) to
obtain the NH$_3$ total columns. A retrieval strategy based on that reported by Dammers et al. (2015) was used, comprising two
microwindows (929.1-931.8 and 963.7-970.0 cm$^{-1}$) to cover the NH$_3$ absorption lines from the $\upsilon_2$ vibrational band in the
mid-IR region. The trace gases H$_2$O, CO$_2$, O$_3$, CO and N$_2$O were taken into account as interfering species for the retrieval for
both stations, and the temperature and pressure profiles were obtained from the US National Centers for Environmental
Prediction (NCEP). Spectroscopic parameters were obtained from the high-resolution transmission molecular absorption
database HITRAN 2008 (Rothman et al., 2009), and the *a priori* profile information about the interfering gases was obtained
from 40-year averages of simulations from the Whole Atmosphere Community Climate Model (WACCM) (Eyring et al.,
2007; Marsh et al., 2013). Since NH$_3$ profiles are not available from WACCM, scaled *a priori* profiles derived from the
global chemical transport model GEOS-Chem v11 NH$_3$ were used instead. Constructed scaled *a priori* profiles have been
used previously from GEOS-Chem simulations for the retrieval of trace gases (Shephard et al., 2011; Shephard and Cady-
Pereira, 2015; Bader et al., 2017). A total of 7992 NH$_3$ columns were retrieved for the UNAM station and 4031 for ALTZ.
The resulting uncertainties obtained with PROFFIT averaged over the entire time series in molecules/cm$^2$ at UNAM were
$1.25 \times 10^{15}$ (random), $8.40 \times 10^{14}$ (systematic), and $1.52 \times 10^{15}$ or 11.50% (total); and $3.09 \times 10^{14}$ (random), $3.14 \times 10^{14}$
(systematic), and $4.42 \times 10^{14}$ or 51.37% (total) at ALTZ. The average degrees of freedom for signal (DOFS) averaged over the
entire time series were 2.03 for UNAM and 1.04 for ALTZ.

## 2.2 IASI-NH$_3$ data product and comparison methodology

IASI measures the infrared thermal radiation emitted by the Earth's surface and the atmosphere from a Sun-synchronous
orbit on board the MetOp platform. Spectra are recorded in the 645-2760 cm$^{-1}$ spectral range at a spectral resolution of 0.5
cm$^{-1}$ (Clerbaux et al., 2009). The instrument crosses the equator at mean local solar times of 09:30 and 21:30, providing
global coverage of the Earth twice a day. IASI has a field-of-view composed of four circular footprints each with a diameter





of 12 km at the nadir view and up to 20 km x 39 km elliptical pixels outside the nadir depending on the satellite viewing angle, complemented by scanning along a swath width of 2200 km off-nadir perpendicular to the ground track (Clarisse et al., 2009; Van Damme et al., 2014, 2015). The IASI-NH₃ retrieval products are based on Artificial Neural Networks (ANNI) that link the Hyperspectral Range Index (HRI, a dimensionless index that represents the amount of NH₃ in the column), to other parameters such as temperature, pressure, and humidity profiles, to derive the NH₃ total column (Van Damme et al.,

2014, 2017; Whitburn et al., 2016). The retrieval scheme does not produce averaging kernels, however, under conditions of high NH₃ and a favourable thermal contrast, IASI has maximum sensitivity to NH₃ in the boundary layer (Clarisse et al., 2010). An error estimate is provided with each individual IASI observation. IASI's average detection limit for NH₃ under large thermal contrast is about 3 ppbv, and can be as low as 1 ppbv under conditions of well-mixed NH₃ throughout a thick boundary layer (Clarisse et al., 2010).

For this study, eleven years of the IASI-A NH₃ total columns (ANNI-NH3-v3) between 2008-2018 were used; details of this version 3 can be found in Appendix A of Van Damme et al. (2021), and was also used by Yamanouchi et al. (2021) and Viatte et al. (2022). The spatial distribution of NH₃ over Mexico City was obtained by averaging all the IASI-A morning observations between January 2008 and December 2018 over this region. The FTIR-NH₃ total columns at UNAM were compared against the IASI-NH₃ total columns over Mexico City to assess the agreement between both data sets. Due to the

high spatiotemporal variability of NH₃, the temporal and spatial coincidence criteria were tested and assessed using the correlations (both $R$ and slope). In addition, as suggested in Dammers et al. (2016), an elevation filter (FTIR station altitude minus IASI observation < 300 m) was applied. The criteria resulting in the best correlations were elevation filter < 300 m, spatial sampling difference < 20 km, maximum temporal sampling difference ± 40 min, and maximum IASI-NH₃ retrieval error of 100%. The seasonal variability comparison and annual averages were performed using only FTIR-NH₃ retrievals

between 9:00 and 10:59 a.m. (Local Time), corresponding to the IASI overpass time over Mexico City, and the < 20 km spatial criterion for IASI-NH₃ total columns. Altzomoni correlation plots with IASI-NH₃ data were not included due to the few coincidences between the FTIR and IASI, a consequence of the station's high-altitude location.

### 2.3 Back-trajectory analysis

To determine the primary sources of NH₃ measured at the UNAM station and to assess the dominant atmospheric transport

pathways during the events with the largest NH₃ columns in the time series, trajectory cluster analysis (Reizer and Orza, 2018) was applied. The 8-hour back-trajectory was selected to capture only air masses traversing the MCMA. Using the UNAM station as the receptor, back-trajectories were calculated using the Hybrid Single-Particle Lagrangian Integrated Trajectory (HYSPLIT) model (Stein et al., 2015, Draxler et al., 1997) at 100 m above the UNAM station level (2280 m.a.s.l.). The 100 m cluster was considered the most representative because NH₃ is more concentrated near the surface.



## 3 Results and Discussion

### 3.1 FTIR-NH₃ time series and temporal variability

The NH$_3$ total column time series retrieved at both FTIR stations are shown in Figure 2. The urban UNAM columns, shown in the top panel, are about one order of magnitude larger than the high-altitude Altzomoni columns. The entire period average NH$_3$ total columns of $1.46 \times 10^{16} \pm 0.64$ molecules/cm² measured at UNAM and $1.87 \times 10^{15} \pm 2.40$ molecules/cm² at Altzomoni are listed and compared to values reported for stations in other parts of the world in Table 1. The Mexico City NH$_3$ total columns are comparable with those reported in Bremen, while they are about twice as large as those measured at Toronto (Canada), Paris (France), and Lauder (New Zealand). Jungfraujoch, a remote high-altitude station in Switzerland with similar characteristics to Altzomoni, presents a significantly lower average NH$_3$ column and also has much lower variability. The reason for this might be that Altzomoni is impacted more frequently by biomass burning events in the dry season and also by the regional boundary layer, receiving polluted air from Mexico City and other large urban centres in the afternoon (Baumgardner et al., 2009).

**Table 1:** Mean NH$_3$ total columns reported from ground-based FTIR stations.

| Station | Location | Time period | Average NH$_3$ total column (molecules/cm$^2$) x $10^{15}$ | Station characteristics | Reference |
|---|---|---|---|---|---|
| Bremen, Germany | 53.10° N, 8.85° E 27 m.a.s.l | 2004-2013 | 13.7 ± 4.24 | Urban, fertilizers, livestock | Dammers et al. (2015) |
| Paris, France | 48.79°N, 2.44°E, 56 m.a.s.l | 2009-2017 | 8.4 ± 8.6 | Urban, surrounding agricultural sources | Tournadre et al., (2020) |
| Jungfraujoch, Switzerland | 46.55°N, 7.98°E, 3580 m.a.s.l | 2004-2013 | 0.18 ± 0.07 | Remote and high altitude, no large sources | Dammers et al. (2015) |
| Toronto, Canada | 43.66°N, 79.40°W, 174 m.a.s.l | 2002-2005 | 5.94 ± 5.14 | Urban, fertilizers, biomass burning | Yamanouchi et al. (2021) |
| | | 2015-2018 | 8.13 ± 7.88 | | |
| UNAM, Mexico | 19.33°N, 99.18°W, 2280 m.a.s.l | 2012-2019 | 14.6 ± 6.39 | Urban, large sources | This study |
| Altzomoni, Mexico | 19.12°N, 98.66°W, 3985 m.a.s.l | 2012-2020 | 1.87 ± 2.40 | Remote and high altitude, no local sources, biomass burning | This study |
| Reunion, Indian Ocean | 20.90°S, 55.5°E, 85 m.a.s.l | 2004-2012 | 0.80 ± 0.54 | Remote, fertilizers, fires | Dammers et al. (2015) |
| Lauder, New Zealand | 45.04°S, 169.68°E, 370 m.a.s.l | 2004-2014 | 4.17 ± 1.40 | Remote, fertilizer, livestock | Dammers et al. (2015) |



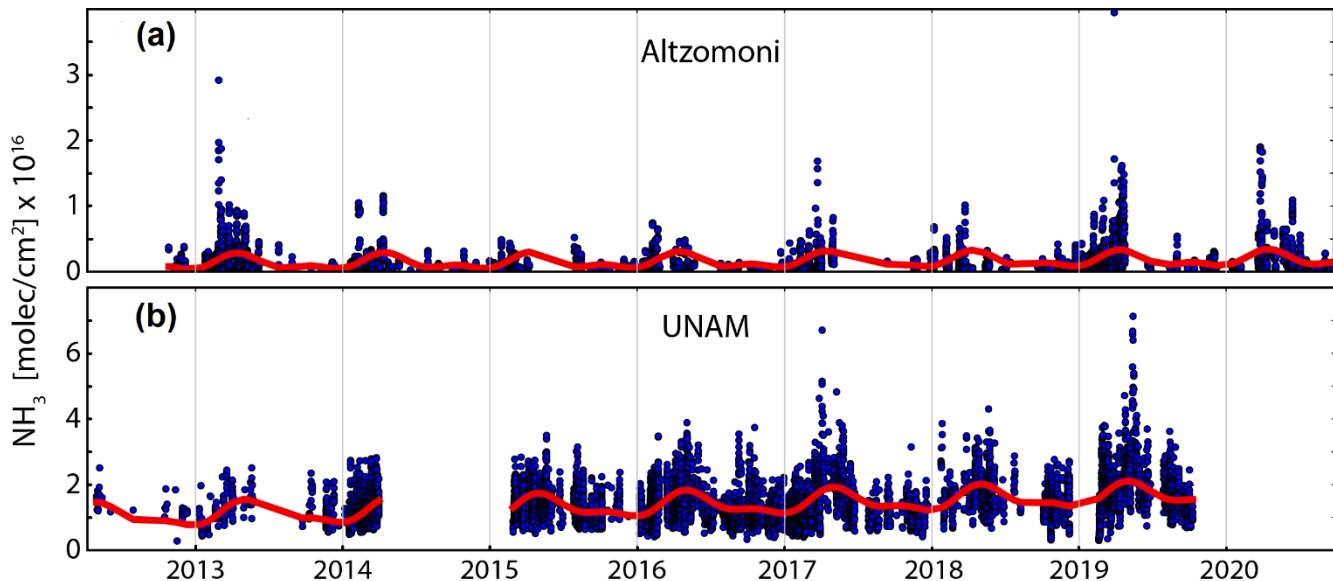

**Figure 2:** Time series of retrieved FTIR-NH₃ data over (a) Altzomoni and (b) UNAM, with a fitted Fourier series (red) to reproduce
seasonality. Note the differences in magnitude.

Figure 2 also shows the fit of a Fourier series (Baylon et. al., 2017) to reproduce seasonality in both stations. A clear cycle is
seen at both stations, with a maximum between mid and late April, and a minimum in late December. However, the
difference between the minimum and the maximum is larger at Altzomoni, with UNAM having greater NH₃ background
concentrations. The average annual increase in the NH₃ columns obtained from the Fourier fit are $92 \pm 3.9 \times 10^{13}$
molecules/cm² yr at UNAM and $8.4 \pm 1.4 \times 10^{13}$ molecules/cm² yr at Altzomoni. Van Damme et al. (2021) reported a trend
from 2008 to 2018 of $2.5 \pm 1.5 \times 10^{13}$ molecules/cm² yr for all of Mexico, which is closer to the Altzomoni value. The
difference in magnitude can be attributed to the datasets and methodology as the present study uses ground-based FTIR
measurements from two sites with higher values in 2019 and 2020, while Van Damme et al. (2021) used IASI satellite data
over a wider region between 2008 to 2018.

The average diurnal variability of the FTIR NH₃ total columns at both stations is displayed in Figure 3. The largest average
NH₃ columns at the urban station are on the order of $1.50 \times 10^{16}$ molecules/cm² and were observed during the morning and
the evening. Although the diurnal pattern is not as evident as in other cities where motor vehicles have been found to be a
dominant source of urban NH₃ (e.g., Osada et al., 2019; Kotnala et al., 2020), traffic emissions in Mexico City still might
play a role in conjunction with other urban sources of atmospheric NH₃. The average NH₃ columns at UNAM have a
minimum of $1.35 \times 10^{16}$ molecules/cm² at 13h, which can be attributed to the conversion to ammonium as was observed by
Moya et al. (2004) that reported the surface gas phase NH₃ and PM NH₄⁺ evolution at an urban site in Mexico City.

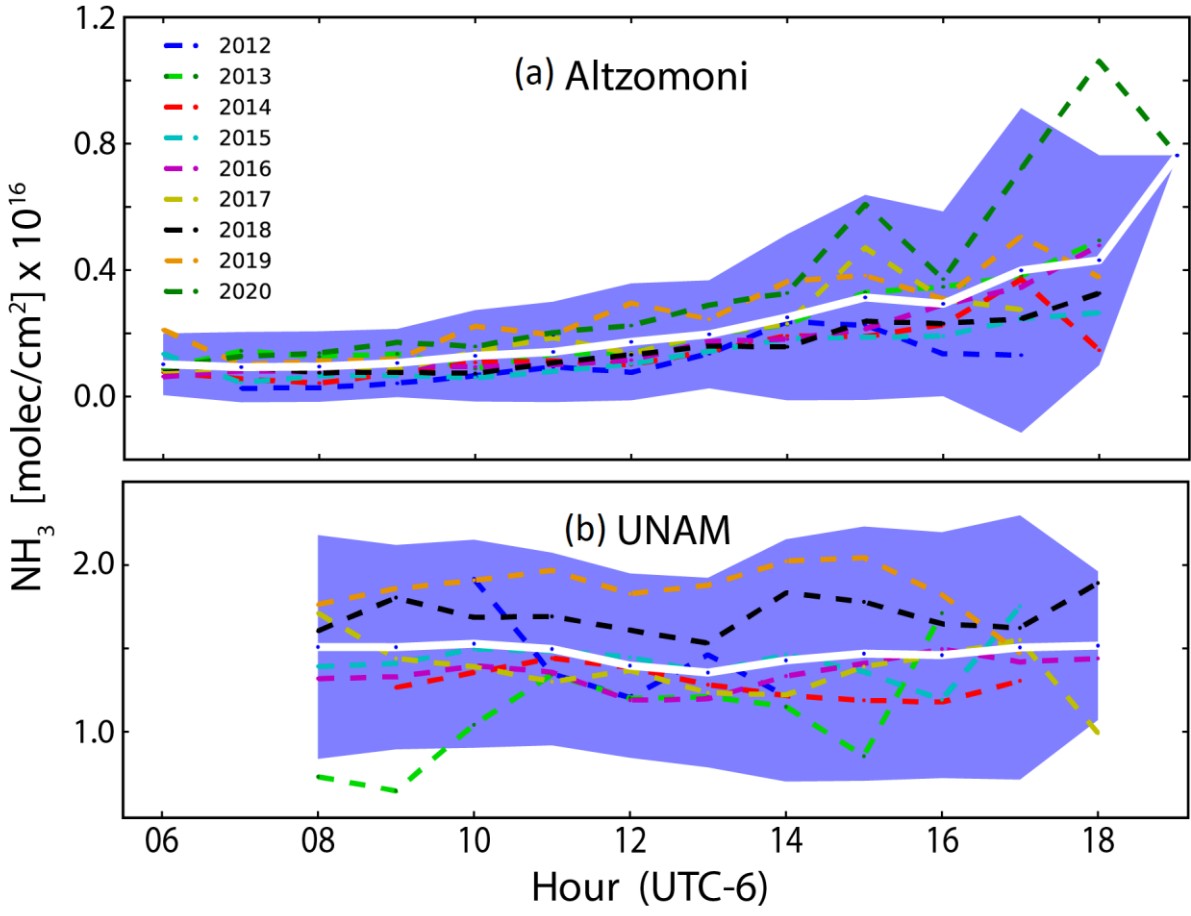

**Figure 3**: Average diurnal evolution of NH₃ total columns over Mexico City at (a) Altzomoni and (b) UNAM. Different years are shown
in different colours, the thick white line is the average for all years, and the blue shading indicates ± 1σ.

The diurnal cycle at the remote station (Figure 3a) is noticeably different, with NH₃ columns that increase systematically as
the day progresses, with the largest variability in the afternoon hours. Altzomoni is located within a natural protected area
with few local sources of NH₃, and even less so during the morning hours when values are around $0.10 \times 10^{16}$
molecules/cm² , when cooler temperatures do not favour the volatilization of NH₃. The columns increase throughout the day,
having the largest average values of $0.76 \times 10^{16}$ molecules/cm² in the evening, probably transported from lower altitudes by
the dynamics of the regional boundary layer (Baumgardner et al., 2009). This is supported by the large variability observed
in the afternoon, since the probability that NH₃ is transported > 1,700 m a.g.l. (above ground level) up to this site strongly
depends on the meteorological conditions, which vary from day to day. Comparisons between daily average NH₃ columns
and the daily averages of some meteorological variables from the RUOA Network (Red Universitaria de Observatorios
Atmosféricos) such as temperature, relative humidity (RH), precipitation, and solar radiation, resulted in weak correlations (r
between 0.1-0.3). However, these correlations were positive with temperature and solar radiation, and negative with RH and



precipitation. The average wind speed for Altzomoni is 4.5 m/s, with dominant winds from the east-southeast and west-northwest, while the average wind speed for UNAM is 1.6 m/s, with dominant winds from the north and the north-northwest.

The seasonal variability of the FTIR $NH_3$ total columns is shown in Figure 4. In general, the pattern is similar at both stations, showing the $NH_3$ temperature dependence with larger $NH_3$ columns in the months of March, April, and May which correspond to the warm-dry season; this season usually has days with clear skies, weak winds, high pressure systems, and biomass burning events (Molina et al., 2020), and also corresponds to the most critical part of the fire season in Mexico City (CENAPRED, 2021; Yokelson et al., 2007). In addition, there are two agricultural seasons in the country, the first one from April to September and the second one from October to March. The fertilizer application combined with meteorological

conditions could favour $NH_3$ volatilization from the agricultural sources contributing to the higher $NH_3$ spring columns observed in Figure 4. Smaller columns are clearly observed during the wet season (June to October), due to the increase in wet deposition, and during the cold-dry season from November to February, due to less favorable conditions for $NH_3$ volatilization. This is in agreement with Viatte et al., (2022). The annual cycle of $NH_3$ columns at the urban UNAM station is similar to that observed at the background Altzomoni station but has a larger amplitude. A study by Sun et al. (2017)

observed that the growing efficiency of three-way catalysts in motor vehicles is responsible for large $NH_3$ emissions detected in urban locations in the USA and China. These emissions are strongly dependent on traffic volume and thus should not have a strong seasonality. On the other hand, emissions originating from agricultural activity usually have a distinct seasonality that depends on the fertilizer application and temperature (Sun et al., 2017; Van Damme et al., 2015, Viatte et al., 2022). In the current study, the annual cycles follows the pattern of temperature (Figure 4c), indicating that emissions from other

sources, such as fires, waste treatment, human or pets emissions, may be contributing significantly to the $NH_3$ detected over the MCMA.

There are more features to note in Figure 4. While Altzomoni and UNAM $NH_3$ columns have similar annual cycles, those at Altzomoni have greater variability throughout the different years during the warm-dry season than during the rest of the year. This might be due to the strong relationship between pollutants reaching the high-altitude station and the boundary

layer dynamics and wind conditions, which are more variable during the warm-dry season. However, another contribution to this variability may be biomass burning activity, which has a maximum during the warm-dry months as has been shown by Cady-Pereira et al. (2017). The 2013 pollution events presented by Cady-Pereira et al. (2017) are seen on April 23, May 9, and May 25 in Figure 2, with May 9 having the largest enhancements at Altzomoni; unfortunately, there are no coincident measurements for UNAM. At Altzomoni, the average $NH_3$ column for May 9, 2013 was $2.39 \pm 0.53 \times 10^{15}$ molecules/cm²,

which is 28% greater than the average column for the entire period (Table 1). However, in 2013, the largest $NH_3$ column measured at Altzomoni was on the evening of February 27 with an average value of $17.4 \pm 0.58 \times 10^{15}$ molecules/cm², almost ten times higher than the average column presented in Table 1. This enhancement on February 27, 2013 seems to be local and of short duration, most likely due to a nearby biomass burning event. This is supported by the detection of active





fires northwest of the site on that date by the MODIS instrument on Aqua as shown on Figure 5. With its high altitude and

few local sources, Altzomoni seems to be more sensitive for the detection of pollution events than UNAM. Even if the fires

are not occurring nearby, the increased lifetimes of emitted pollutants at these altitudes may favour transport over longer

distances to Altzomoni.

**Figure 4:** Monthly averages showing annual cycle of NH₃ over Mexico City at (a) Altzomoni and (b) UNAM. The thick white line is the
average for all years and the shading indicates ± 1σ. (c) Monthly averages of temperature at both stations between 2014-2018 for
Altzomoni and 2012-2018 for UNAM. The shaded area indicates ± 1σ.





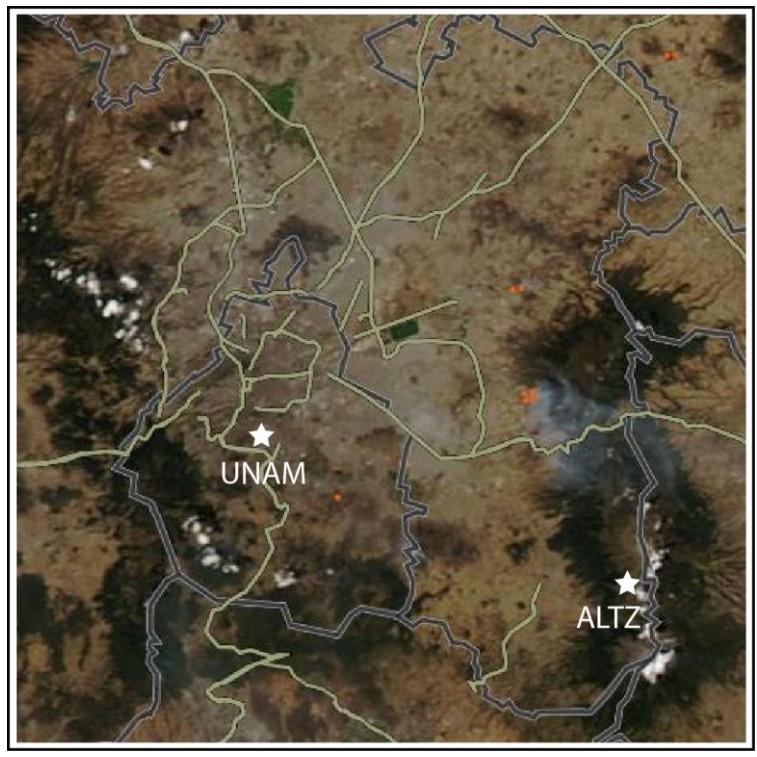

**Figure 5:** Snapshot of Mexico's City fire events on February 27, 2013, from the Aqua MODIS instrument with true colour corrected reflectance and resolution of 250 m, obtained from NASA Worldview Snapshots (https://earthdata.nasa.gov). The red dots show the fire events. The stars indicate the location of the UNAM and Altzomoni stations.

### 3.2 Satellite observations: comparison with ground-based measurements and spatial distribution

The correlation between IASI-NH$_3$ and the ground-based FTIR-NH$_3$ total columns at UNAM from Nov 2013 to Dec 2018 is shown in Figure 6. The coincidence criteria are described in Sect. 2.2, and are based on values used in previous validations of IASI products using ground-based FTIR data (Dammers et al., 2016), including an elevation correction using the Space Shuttle Radar Topography Mission Global Product at 3 arcsec resolution over Mexico City (SRTMGL3, Farr et al., 2007). A total of 64 coincident data pairs were found, from which a correlation coefficient $R = 0.72$ and a mean relative difference (MRD) of -32.2 ± 27.5 % were obtained (± 1σ). These results are consistent with $R = 0.64$ and MRD = -30.8 ± 43.9 % reported by Dammers et al. (2016) for this region using an older version of the IASI-NH$_3$ product. The correlation is also similar to that of Tournadre et al. (2020), who obtained R= 0.79, when comparing IASI-NH$_3$ to FTIR-NH$_3$ columns using a similar instrument (Vertex 80) in Paris, and to the $R = 0.80$ and MRD = -32.4 ± 56.3 % reported for 547 coincidences from several ground-based FTIR stations and IASI-NH$_3$ (Dammers et. al., 2016). The IASI ANNI-NH$_3$-v3 product is thus in agreement with the ground-based data and even presents an improved correlation compared to the previous result. However, an underestimation in the IASI-NH$_3$ total columns of approximately 32% over Mexico City persists, for reasons that need to be investigated further.





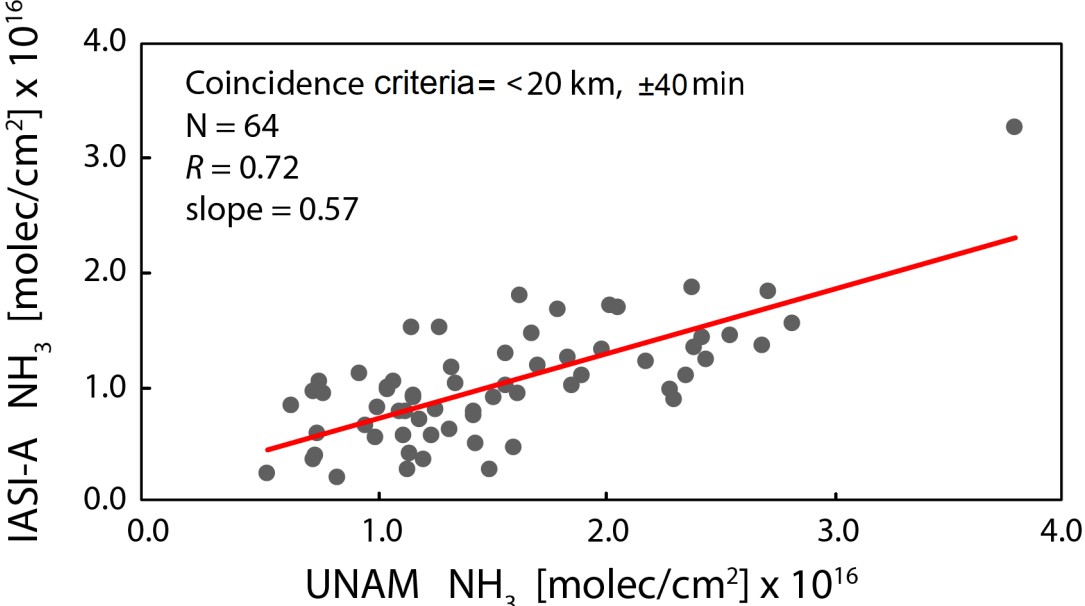


**Figure 6:** Correlation plot for IASI-A NH$_3$ vs. UNAM FTIR-NH$_3$ total columns, with coincidence criteria of < 20 km, ± 40 minutes, elevation (FTIR station – IASI observation) < 300 m, IASI-NH$_3$ retrieval error < 100% between Nov 2013 and Dec 2018.

The spatial distribution of NH$_3$ over MCMA as observed by IASI is presented in Figure 7. The distribution shows a clear NH$_3$ enhancement in the northeast section of Mexico City and in part of the Estado de México region. There are several

potentially important sources located in this area: Mexico City International Airport, an area of continuous traffic emissions; the Bordo Poniente compost plant, which treats around 1500 tons of daily organic waste from the city; wastewater discharge and treatment bodies with nearby bird colonies such as the regulation lagoon Cola de Pato; and the agricultural area at Texcoco. The combination of these factors, along with the high population density of this area, are likely to be cause of the NH$_3$ enhancements observed in this part of the city. This enhancement is partially in agreement with the location of the larger

NH$_3$ emissions reported in the Mexico City Emissions Inventory of 2016 (SEDEMA, 2018) shown in Figure 1 in the northern part of the city, which is mainly associated with population activities and domestic animals' excreta. Figure 7d will be discussed in Section 3.3.

Figures 7a-c show the variations of NH$_3$ over the year, with the largest columns measured during the warm-dry season and to the northeast of UNAM. In contrast, the NH$_3$ columns are reduced in the wet season when wet deposition can occur, and are

smallest during the cold-dry season, when there are lower temperatures and less NH$_3$ volatilization. To investigate the influence of local topography on the NH$_3$ distribution, Figure 8 compares the average IASI-NH$_3$ total column spatial distribution (a) with altitude (b). The figure illustrates that the highest columns are located at the lowest altitudes while the lowest columns are at higher altitudes, reflecting the source locations and the boundary layer dynamics. The figure shows that the main NH$_3$ sources in MCMA are located in the most urbanized areas in Mexico City and Estado de Mexico at an

altitude of around 2250 m. These urban emissions agree with the statement of Li et. al. (2020) that human NH$_3$ emissions





contributing significantly to the total NH$_3$ emissions in hot and highly populated urban areas such as Mexico City. A rough estimation using 25 °C as an average diurnal temperature for Mexico City and the 0.4 mg of NH$_3$ per hour at 25 °C from Li et. al. (2020) resulted in an estimate of 34 tonnes of NH$_3$ per year, a contribution of the same order of magnitude as all the "point sources" and "urban waste" combined according to the Mexico City Emissions Inventory for 2016 (SEDEMA, 2018).

**Figure 7:** Spatial distribution of IASI-A NH$_3$ total columns for the morning overpass of MetOp-A over Mexico City, averaged over 2008-2018 for: (a) the cold-dry season (November – February), (b) the warm-dry season (March-May), and (c) the wet season (June-October). HYSPLIT back-trajectories for the UNAM site are shown in panel (d). The black stars indicate the locations of both stations, and Mexico City International Airport is shown for reference.




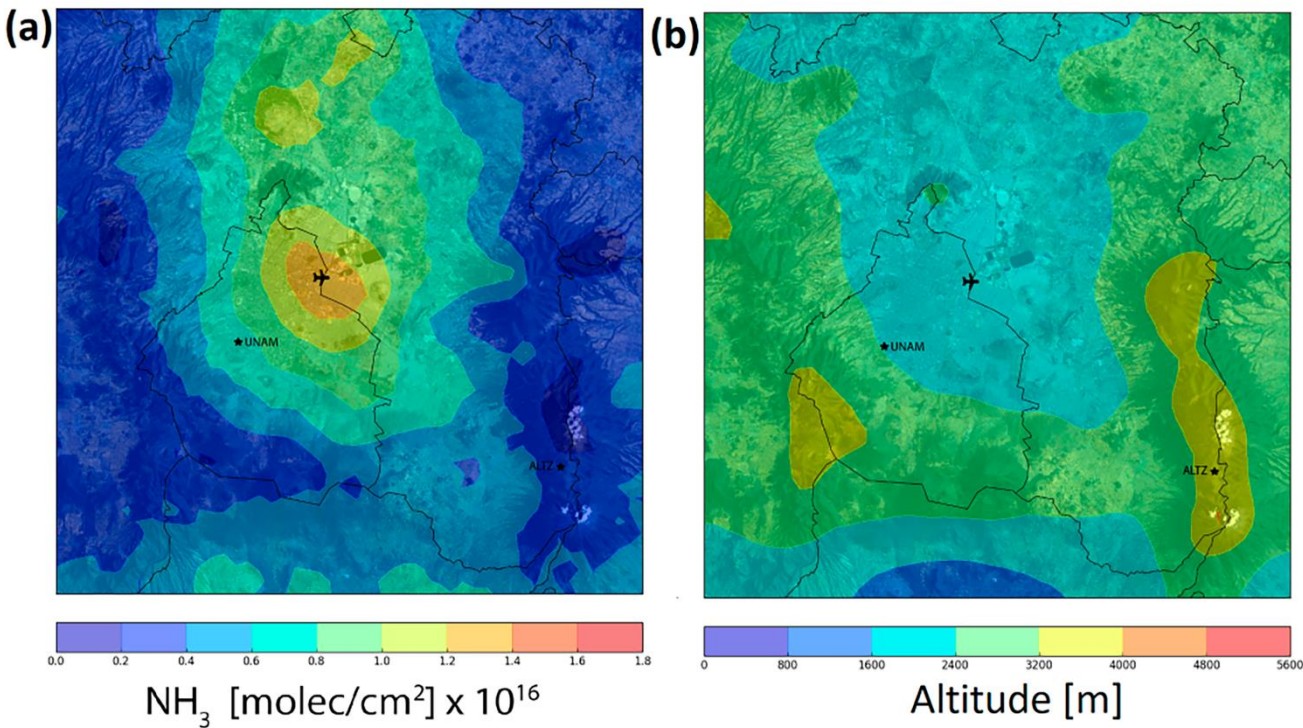

**Figure 8:** Spatial distribution of (a) IASI-A NH$_3$ total columns for the morning overpass of MetOp-A over Mexico City, averaged over 2008-2018 (b) altitudes in Mexico City from the Space Shuttle Radar Topography Mission Global product City (SRTMGL3, Farr et al., 2007). The black stars indicate the location of ground-based stations, and Mexico City International Airport is shown for reference.

Comparisons between the NH$_3$ seasonal variability and time evolution over the UNAM station in Mexico City were performed using morning (9-11 h) FTIR-NH$_3$ columns and IASI-NH$_3$ columns with a spatial criterion of < 20 km from the UNAM station (Figure 9a). The NH$_3$ seasonal variability over Mexico City in Figure 7a, is similar for both the IASI and ground-based FTIR NH$_3$ columns and is in agreement with Figure 4. However, the IASI-NH$_3$ shown a consistent negative bias. The time evolution is represented with the IASI-NH$_3$ and FTIR-NH$_3$ annual averages in Figure 9b. The datasets suggest an increasing trend in the annual averages of the NH$_3$ total columns, with larger columns observed in the most recent years, even in Altzomoni except for 2013, which was affected by the February 27 event as discussed previously. Comparing the average IASI-NH$_3$ columns (using the < 20 km spatial criteria) for 2008 (7.03 x 10$^{15}$ ± 3.41 x 10$^{15}$ molecules/cm²) vs. 2018 (11.4 x 10$^{15}$ ± 6.64 x 10$^{15}$ molecules/cm²), there is an increment of 62 % over a decade for Mexico City. This positive increment agrees with the trend of 8.4 ± 5.2 % 10 yr$^{-1}$ reported by Van Damme et al. (2021) for all of Mexico.







**Figure 9:** IASI and FTIR NH₃ total columns with coincidence criteria of < 20 km and < 2 hr: (a) monthly averages at UNAM, (b) diurnal annual averages at UNAM and Altzomoni. The shaded area indicates ±1σ.



### 3.3 Back-trajectory analysis and origin of observed NH₃ in Mexico City

A cluster analysis was applied using 8-hour back-trajectories 100 m above UNAM station to identify the main transport pathways for air masses arriving at UNAM station that correspond to the highest average hourly $NH_3$ total columns (Figure

7d). The Total Spatial Variance (TSV) method (Draxler et al., 2021) was used to fit the number of clusters, which was set to three in order to represent the primary trajectories. It was found that 68% of the trajectories originate from the north (red line with black dots), 27% from the west-southwest (green line), and 5 % from the south (blue line). However, by looking at the individual back-trajectories of the red cluster (the black-thin lines in Figure 7d), it is seen that most of the $NH_3$ detected at UNAM comes from a variety of local sources and does not suggest that $NH_3$-enriched air masses are being transported to the

urban station only from the enhancement region to the northeast observed in Figures 7a to 7c. This is in agreement with Viatte et al. (2022). The relationship between the back-trajectories and measured $NH_3$ columns can be explained by the fact that Mexico City is located in a basin; the wind fields are constricted in this basin and in general they are breeze winds (6 km/h). Under these conditions, small locally distributed $NH_3$ urban emissions seem to be the main cause of the high column values of this pollutant measured at the UNAM station, this agrees with Figure 8 where the main $NH_3$ sources in MCMA are

seen to be urban.

### 3 Conclusions

This work presented the temporal and spatial distribution of $NH_3$ total columns over the Mexico City Metropolitan Area derived from two ground-based FTIR spectrometers and IASI satellite observations. The average $NH_3$ total column at the urban UNAM site ($1.46 \pm 0.64$ x $10^{16}$ molecules/cm²) is considerably higher than that at the remote station Altzomoni ($1.87$

$\pm 2.40$ x $10^{15}$ molecules/cm²), with a clear difference in the diurnal cycle but similar seasonal variability. $NH_3$ spatial distribution from IASI shows the highest $NH_3$ columns in the northeast part of the city, an area surrounded by water bodies, a landfill, a compost plant for the treatment of all the organic waste, and the airport. The IASI ANNI-NH₃-v3 data product underestimates the $NH_3$ total columns over Mexico City, with a mean relative difference of 32%, over the period 2008-2018 but showed a similar temporal variability and a good correlation with FTIR measurements ($R$=0.72). The analysis of back-

trajectories for the largest $NH_3$ enhancement events suggests that most of the $NH_3$ measured at the urban station is coming from local sources. The $NH_3$ observed at the remote site is most likely transported from the surroundings and it is influenced by biomass burning events. These results present evidence that sources other than from agriculture, such as motor vehicles, fires, human emissions, domestic animals, water discharge, and waste, have a significant contribution to the total $NH_3$ budget in the city. In general, an average annual increase is observed in Mexico City from both ground-based stations ($92 \pm 3.9$ x

$10^{13}$ molecules/cm² yr at UNAM, $8.4 \pm 1.4$ x $10^{13}$ molecules/cm² yr at Altzomoni) and IASI (62 % / 10 yr⁻¹). A complementary study using surface $NH_3$ and PM concentrations from passive samplers and microsensors around this region is in progress. These observations, together with model data, will examine the role of reactive nitrogen in the pollution of Mexico City. A revaluation of $NH_3$ emission sources contribution in the Mexico City inventory is suggested. Measures to





mitigate NH$_3$ emissions and reduce these positive trends are important, given that NH$_3$ is closely linked to secondary aerosol

formation and the deterioration of ecosystems.

*Data availability:* The UNAM and Alzomoni FTIR data used in this study are available from the correspondent author upon request (beatriz.herrera@mail.utoronto.ca). The meteorological data for both FTIR stations are available at https://www.ruoa.unam.mx/. The IASI ANNI-NH$_3$-v3 L2 data can be access at https://iasi.aeris-data.fr/NH3/. MODIS active fire snapshots are available at https://wvs.earthdata.nasa.gov/. The code of the HYSPLIT model can be obtained from

https://www.arl.noaa.gov/hysplit/getrun-hysplit/ . Last access to all URLs: 11 March 2022.

*Competing interests:* The authors declare that they have no conflict of interest.

**Author contributions**

BH is the main author of the paper, analyzed the data, made most of the figures, and wrote the text. WS contributed to the data analysis. MG contributed to create some figures. AM contributed with the back-trajectory analysis. MG, KS, CR, FH,

TB, CV, ED, LC, and MVD contributed to the manuscript writing and provided support. AB provided technical support. ED provided the Space Shuttle Radar Topography Mission Global product and support with the NH$_3$ retrievals. LC and MVD developed the IASI-NH3 product. CV and MVD provided the IASI data. All authors reviewed the manuscript.

**Acknowledgements**

We acknowledge the use of imagery from the Worldview Snapshots application (https://wvs.earthdata.nasa.gov), part of the

Earth Observing System Data and Information System (EOSDIS). BH acknowledges CONACYT for the scholarship granted. The RUOA Network (www.ruoa.unam.mx) is acknowledged for making the meteorological measurements available. IASI is a joint mission of EUMETSAT and the Centre National d'Etudes Spatiales (CNES, France). The authors acknowledge the ULB-LATMOS team for providing the IASI data, and for the development of the retrieval algorithms.

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
