# Peer review of "Measurement report: Evolution and distribution of NH3 over Mexico City from ground-based and satellite infrared spectroscopic measurements"

_Atmospheric Chemistry and Physics, 2022_

## Referee Comment (RC1)

**Review of "Measurement report: Evolution and distribution of NH3 over Mexico City from ground-based and satellite infrared spectroscopic measurements"**

**Summary**

This paper describes a study of the changes in NH3 over Mexico City as observed by  FTIR instruments at two ground stations, one in an urban area and one in a more remote location, and from the IASI instrument. The FTIR and IASI data both showed similar seasonal variability, peaking in April and May, and a significant increase in NH3 amounts over the observing  period. Interestingly the largest NH3 amounts are measured in the northeastern corner of the MCMA and appear to have local sources, as predicted rom an emissions inventory and confirmed by a back trajectory analysis.

The paper is well laid out and clearly written. The plots are very high quality and easily understood. It requires only some minor edits and additions to be acceptable for publication.

**Technical comments**

Line 132: Please provide a little more detail on the a priori profiles. Are there more than one? If yes, how are they chosen?

Line 279: Could the authors propose some possible explanations for the column underestimation by IASI?

**Minor edits**

Line 71: come from

Line 74: "The inventory also strongly attributes the NH3 sources to a range of population activities and
75 feces from domesticated animals". This sentence is not clear.

Line 89: …all of which are classified as …

Line 152: …when the thermal contrast is large

Line 171: Please clarify this sentence: "The 8-hour back-trajectory was selected to capture only air masses traversing the MCMA".

Line 174: …NH3 is mostly concentrated near the surface

Line 177: The average NH3 total columns for the entire period ( 1.46x1016 ± 0.64 molecules/cm2 at UNAM and 1.87x1015 ± 2.40 molecules/cm2  at  Altzomoni) are listed and …

Line 207: …attributed to the conversion to ammonium, as was observed by
Moya et al. (2004) when describing the evolution of the surface gas phase NH3 and PM NH4+ evolution at an urban site in Mexico City.

Line 314: the evolution with time…

Line 316:  … in Figure 9a

Line 317: However, IASI-NH3 shows a consistent negative
bias. The evolution with time is is represented by the IASI-NH3 and FTIR-NH3 annual averages in Figure 9b.

Line 320: even in Altzomoni,

Line 322: … there is an increase of 62 % over a decade for Mexico City, in agreement with the trend ….

Line 329: … at this station

Line 334: from a variety of local sources and does not show only the transport of  NH3-enriched air masses from the enhancement region to the northeast observed in Figures 7a to 7c. This is in agreement with
Viatte et al. (2022).

Line 339: This sentence is not clear or does not follow: this agrees with Figure 8 where the main NH3 sources in MCMA are seen to be urban.

---

## Referee Comment (RC3)

**Review of "Measurement report: Evolution and distribution of NH3 over Mexico City from ground-based and satellite infrared spectroscopic measurements"**

**General Comments**

This study aims to constrain the spatial and temporal distribution of total ammonia columns in the Mexico City Metropolitan Area (MCMA) using ground-based FTIR spectrometers (from two sites) and satellite-based IASI observations. The study finds meaningful differences between the two surface sites (particularly in the diurnal cycle), illustrates the spatial heterogeneity in NH3 column concentrations within the MCMA using IASI data, and highlights annual NH3 trends across both datasets. The study also uses a back-trajectory cluster analysis to suggest a large portion of the measured NH3 is local and urban in nature, with significant contributions from non-agricultural sources.

The study is well-written and highlights important conclusions, namely the need for a re-evaluation of NH3 sources in the MCMA region. It is a useful contribution to the literature and provides a basis for more targeted work in this region. That said, the paper provides a somewhat limited discussion of a few key methods and conclusions that are central to its interpretation. With this in mind, I provide the following comments below and recommend that these issues be addressed prior to publication in ACP.

**Specific Comments**

**Impact of Human Emissions**

The authors allude to the importance of human emissions in the region (even providing a back-of-the envelope estimate). However, based on this very estimate, human emissions appear to be 3 orders of magnitude lower than the total regional NH3 flux. It thus seems unlikely that these sources are 'meaningful', but they are presented in this study as being an important underappreciated source. The authors also discuss the importance of local and urban sources multiple times in the study, citing human emissions as one of these sources. However, given the relatively small contribution of human emissions to the total flux, it stands to reason that there are other 'main' urban and local sources that should be prioritized over human emissions in order to develop accurate inventories. It would thus be a very useful contribution if the authors could quantitatively or qualitatively prioritize the relative importance of these underappreciated sources (fires, waste treatment, humans, pets, etc.) and discuss how their estimates differ from the current inventory estimates.

**IASI retrieval and validation**

Since the IASI retrieval scheme does not produce averaging kernels, could the authors expand the discussion in Section 2.2 to include a more in-depth overview of the underlying uncertainties and sensitivity constraints in the retrieval?

Could the authors also please provide a more detailed description of the ANNI that estimates NH3 column concentrations based on the various input parameters (e.g., a list of all the parameters considered, the uncertainties associated with each parameter, the uncertainties associated with the ANNI transformation itself). A brief discussion in the main-text of the differences in the dynamic range of the data used to train the ANNI vs. the data inputs in this study would also be helpful.

Could the authors also please build on the above request to expand on potential reasons for the IASI column underestimation relative to the surface instruments? This is important in order to appropriately interpret the broader IASI spatial patterns.

**Back-trajectory analysis**

Please briefly expand Section 2.3 and Section 3.3 to provide more information on the specifics of the back-trajectory cluster analysis such that it can be reproducible by a third-party reader of this paper. Please also provide a brief discussion on how sensitive your results were based on the number of clusters assumed, clustering technique, etc.

**Minor Comments**

Line 71: 'come from'

Line 74: Suggest restructuring this sentence to – 'The inventory also attributes a meaningful NH3 contribution (%) to a range of different population activities (e.g., X,Y,Z) and feces from domesticated animals'

Line 89: '.. , all classified as ..'

Line 178: 'The entire period …' – Please restructure this sentence

Line 206: '.. conversion to ammonium, as was observed …'

Line 314: 'Comparisons between the seasonal and temporal variability of NH3 over …'

Line 317: 'However, IASI-NH3 shows a …'

Line 318: 'The temporal evolution is represented using ..'

Line 320: 'even in Altzomoni, except for 2013 which was …'

Line 321: 'averaged'

Line 322: 'molecules/cm2), yields a 62% increase in Mexico City over the course of a decade, in agreement with the trend of …..'

Line 332: Please restructure for clarity

---

## Author Comment (AC1)

**RC1 of "Measurement report: Evolution and distribution of NH3 over Mexico City from ground-based and satellite infrared spectroscopic measurements"**

**Reviewer 1**

**Summary**

This paper describes a study of the changes in NH3 over Mexico City as observed by FTIR instruments at two ground stations, one in an urban area and one in a more remote location, and from the IASI instrument. The FTIR and IASI data both showed similar seasonal variability, peaking in April and May, and a significant increase in NH3 amounts over the observing period. Interestingly the largest NH3 amounts are measured in the northeastern corner of the MCMA and appear to have local sources, as predicted rom an emissions inventory and confirmed by a back trajectory analysis. The paper is well laid out and clearly written. The plots are very high quality and easily understood. It requires only some minor edits and additions to be acceptable for publication.

Response: The authors would like to thank the reviewer for this positive review and all the suggestions to improve this work. We addressed each comment below and in the revised manuscript.

**Technical comments**

Line 132: Please provide a little more detail on the a priori profiles. Are there more than one? If yes, how are they chosen?

Response: The a priori profiles were derived from five years of averaged $NH_3$ simulations from the global chemical transport model GEOS-Chem v11. We include this information in the revised manuscript as follows: "scaled *a priori* profiles derived from 5-years of averaged $NH_3$ simulations from the global chemical transport model GEOS-Chem v11 were used instead".

Line 279: Could the authors propose some possible explanations for the column underestimation by IASI?

Response: The underestimation could be attributed to a combination of more randomly distributed error sources and large systematic errors, as was attributed in Dammers et al., (2017), using an older version of the IASI ANNI-$NH_3$ product. We include this information in the revised manuscript as follows: "Dammers et al. (2017), using an older version of the IASI ANNI-$NH_3$ product, attributed these differences to a combination of more randomly distributed error sources and large systematic errors, however, these reasons need to be investigated further."

**Minor edits**

Line 71: come from

Response: Done

Line 74: "The inventory also strongly attributes the NH3 sources to a range of population activities and feces from domesticated animals". This sentence is not clear.

Response: We have modified this sentence in the revised manuscript as: "The inventory strongly attributes domestic emissions of $NH_3$ to feces from domesticated animals."

Line 89: …all of which are classified as …

Response: Done

Line 152: …when the thermal contrast is large

Response: Done

Line 171: Please clarify this sentence: "The 8-hour back-trajectory was selected to capture only air masses traversing the MCMA".

Response: We have modified this sentence in the revised manuscript as: "Eight-hour back-trajectories were selected to capture the air masses passing over the MCMA."

Line 174: …NH3 is mostly concentrated near the surface

Response: Done

Line 177: The average NH3 total columns for the entire period ( $1.46 \times 10^{16} \pm 0.64$ molecules/cm2 at UNAM and $1.87 \times 10^{15} \pm 2.40$ molecules/cm2 at Altzomoni) are listed and …

Response: Done

Line 207: …attributed to the conversion to ammonium, as was observed by Moya et al. (2004) when describing the evolution of the surface gas phase NH3 and PM NH4+ evolution at an urban site in Mexico City.

Response: Done

Line 314: the evolution with time…

Response: We have modified this sentence to also complement RC3 in the revised manuscript as "Comparisons between the seasonal and temporal variability of $NH_3$ over the …"

Line 316: … in Figure 9a

Response: Done

Line 317: However, IASI-NH3 shows a consistent negative bias. The evolution with time is is represented by the IASI-NH3 and FTIR-NH3 annual averages in Figure 9b.

Response: Done

Line 320: even in Altzomoni,

Response: Done

Line 322: … there is an increase of 62 % over a decade for Mexico City, in agreement with the trend ….

Response: Done and complemented with response to RC3

Line 329: … at this station

Response: Done

Line 334: from a variety of local sources and does not show only the transport of NH3-enriched air masses from the enhancement region to the northeast observed in Figures 7a to 7c. This is in agreement with Viatte et al. (2022).

Response: Done

Line 339: This sentence is not clear or does not follow: this agrees with Figure 8 where the main NH3 sources in MCMA are seen to be urban

Response: We have modified this sentence in the revised manuscript as: "this agrees with Figure 8 which shows that the main NH$_3$ sources in MCMA seem to be urban."

Extra:

Line 116:  We added a line and a reference

"The first study presenting and validating the combined usage of trace gas products obtained from dedicated retrievals with FTIR spectra measured at UNAM and ALTZ was that of Plaza-Medina et al. (2017)."

Reference:

Plaza-Medina, E. F., Stremme, W., Bezanilla, A., Grutter, M., Schneider, M., Hase, F., and Blumenstock, T.: Ground-based remote sensing of O$_3$ by high- and medium-resolution FTIR spectrometers over the Mexico City basin, Atmos. Meas. Tech., 10, 2703-2725, https://doi.org/10.5194/amt-10-2703-2017, 2017.

We update the Viatte et al. (2022) reference as:

Viatte, C., Abeed, R., Yamanouchi, S., Porter, W., Safieddine, S., Van Damme, M., Clarisse, L., Herrera, B., Grutter, M., Coheur, P.-F., Strong, K., and Clerbaux, C.: NH3 spatio-temporal variability over Paris, Mexico and Toronto and its link to PM2.5 during pollution events, EGUsphere [preprint], https://doi.org/10.5194/egusphere-2022-413, 2022.

---

## Author Comment (AC2)

**RC2 of "Measurement report: Evolution and distribution of NH3 over Mexico City from ground-based and satellite infrared spectroscopic measurements"**

**Reviewer 1**

Please note that the x-axis labels on Figure 7 are not correct.

Response: Thanks so much for pointing out this issue. We corrected the labels.

---

## Author Comment (AC3)

**RC3 of "Measurement report: Evolution and distribution of NH3 over Mexico City from ground-based and satellite infrared spectroscopic measurements"**

**Reviewer 2**

**General Comments**

This study aims to constrain the spatial and temporal distribution of total ammonia columns in the Mexico City Metropolitan Area (MCMA) using ground-based FTIR spectrometers (from two sites) and satellite-based IASI observations. The study finds meaningful differences between the two surface sites (particularly in the diurnal cycle), illustrates the spatial heterogeneity in NH3 column concentrations within the MCMA using IASI data, and highlights annual NH3 trends across both datasets. The study also uses a back-trajectory cluster analysis to suggest a large portion of the measured NH3 is local and urban in nature, with significant contributions from non-agricultural sources. The study is well-written and highlights important conclusions, namely the need for a re-evaluation of NH3 sources in the MCMA region. It is a useful contribution to the literature and provides a basis for more targeted work in this region. That said, the paper provides a somewhat limited discussion of a few key methods and conclusions that are central to its interpretation. With this in mind, I provide the following comments below and recommend that these issues be addressed prior to publication in ACP.

The authors would like to thank the reviewer for this positive review and all the suggestions to improve this work. We address each comment below and in the revised manuscript.

**Specific Comments**

**Impact of Human Emissions**

The authors allude to the importance of human emissions in the region (even providing a back-of-the envelope estimate). However, based on this very estimate, human emissions appear to be 3 orders of magnitude lower than the total regional NH3 flux. It thus seems unlikely that these sources are 'meaningful', but they are presented in this study as being an important underappreciated source. The authors also discuss the importance of local and urban sources multiple times in the study, citing human emissions as one of these sources. However, given the relatively small contribution of human emissions to the total flux, it stands to reason that there are other 'main' urban and local sources that should be prioritized over human emissions in order to develop accurate inventories. It would thus be a very useful contribution if the authors could quantitatively or qualitatively prioritize the relative importance of these underappreciated sources (fires, waste treatment, humans, pets, etc.) and discuss how their estimates differ from the current inventory estimates.

Response: We found an update of the inventory for 2018 (SEDEMA, 2021) and there were some changes in the different sources, with the main difference in the "area sources". They reported 46931 tonnes of $NH_3$/yr, with 0.3% of $NH_3$ emissions in Mexico City come from "point sources" such as industry, 5.5% from "mobile sources" such as vehicles, and 94.2% from "area sources" including urban waste (1.09%), agriculture (9.44%), livestock (13.92%), domestic emissions (69.73%), and controlled fires (0.01%). As described in the previous sentence, the domestic emissions category was implemented as the most significant contributor to $NH_3$ sources in Mexico City, and the percentage attributed to agriculture and livestock also increase significantly, in agreement with this work. Given the information presented in this measurement report, it is not possible to specifically and quantitatively assign the relative importance of

each one of the underappreciated sources, however, as the updated emissions inventory presented, domestic emissions, including the emissions from domesticated animals, are a significant contributor to $NH_3$ and are becoming more relevant as can be observed in the pie chart below. In Mexico, the estimated number of dogs and cats in the whole country is around 23 million and 70% are homeless (GACETA: LXIV/1PPO-56/86584, 2022). The authors believed that the fires are another $NH_3$ source that is still underestimated in the inventory, as was shown with the Altzomoni enhancement. Further improvements in the estimated emissions are still needed.

[Figure]

We implemented this new information in the revised manuscript by updating the percentages as well as the references below.

GACETA: LXIV/1PPO-56/86584: https://www.senado.gob.mx/64/gaceta_del_senado/documento/86584, last access: 2 September 2022.

SEDEMA Secretaría del Medio Ambiente de la Ciudad de México: Inventario de Emisiones de la Zona Metropolitana del Valle de México 2018. Dirección General de Gestión de la Calidad del Aire, Dirección de Proyectos de Calidad del Aire. Ciudad de México. URL: http://www.aire.cdmx.gob.mx/descargas/publicaciones/flippingbook/memoria-inventario-emisiones-2018/memoria_inventario_emisiones_2018.pdf, 2021.

**IASI retrieval and validation**

Since the IASI retrieval scheme does not produce averaging kernels, could the authors expand the discussion in Section 2.2 to include a more in-depth overview of the underlying uncertainties and sensitivity constraints in the retrieval? Could the authors also please provide a more detailed description of the ANNI that estimates NH3 column concentrations based on the various input parameters (e.g., a list of all the parameters considered, the uncertainties associated with each parameter, the uncertainties associated with the ANNI transformation itself). A brief discussion in the main-text of the differences in the dynamic range of the data used to train the ANNI vs. the data inputs in this study would also be helpful. Could the authors also please build o n the above request to expand on potential reasons for the IASI

column underestimation relative to the surface instruments? This is important in order to appropriately interpret the broader IASI spatial patterns.

Response: We expanded Section 2.2 by adding more information regarding the ANNI-NH$_3$ retrieval process, including more details about the input parameters, sensitivity, uncertainty, and previous comparisons with ground-based FTIR measurements. The potential reasons for the IASI column underestimation were addressed in the response for RC1 and are also included in the revised manuscript as follows:

"The IASI-NH$_3$ retrieval products are based on Artificial Neural Networks (ANNI) that link the Hyperspectral Range Index (HRI), a calculated dimensionless index that represents the amount of NH$_3$ in the column, to other input parameters such as temperature, pressure, water vapor profiles, and parametrized vertical profiles of NH$_3$ to derive the NH$_3$ total column. The algorithm maps the HRI to the NH$_3$ total column using a trained neural network; the uncertainty of each NH$_3$ column can be estimated by the propagation of the input parameters' uncertainties. However, large HRI values (more than 3 σ) are associated to a confident detection of NH$_3$ (Van Damme et al., 2014, 2017; Whitburn et al., 2016). The current spectral range for the retrieval process is set to 812-1126 cm$^{-1}$ to increase the sensitivity of NH$_3$ and reduce interferences (Van Damme et al., 2021). The retrieval scheme does not produce averaging kernels, however, previous studies comparing the IASI-NH$_3$ product with ground-based FTIR measurements have demonstrated good agreement (e.g., Dammers et al., 2016,2017; Lutsch et al., 2019; Tournadre et al., 2020; Yamanouchi et at., 2020). In addition, under conditions of high NH$_3$ and when the  thermal contrast is large, IASI has maximum sensitivity to NH$_3$ in the boundary layer (Clarisse et al., 2010). An error estimate is provided with each individual IASI observation; for this work IASI observations with errors less than 100% were used. IASI's average detection limit for NH$_3$ under large thermal contrast is about 3 ppbv, and can be as low as 1 ppbv under conditions of well-mixed NH$_3$ throughout a thick boundary layer (Clarisse et al., 2010)."

We added the reference below.

Lutsch, E., Strong, K., Jones, D. B. A., Ortega, I., Hannigan, J. W., Dammers, E., Shephard, M. W., Morris, E., Murphy, K., Evans, M. J., Parrington, M., Whitburn, S., Van Damme, M., Clarisse, L., Coheur, P. F., Clerbaux, C., Croft, B., Martin, R. V., Pierce, J. R., and Fisher, J. A.: Unprecedented Atmospheric Ammonia Concentrations Detected in the High Arctic From the 2017 Canadian Wildfires, J. Geophys. Res. Atmos., 124, 8178–8202, https://doi.org/10.1029/2019JD030419, 2019.

**Back-trajectory analysis**

Please briefly expand Section 2.3 and Section 3.3 to provide more information on the specifics of the back-trajectory cluster analysis such that it can be reproducible by a third-party reader of this paper. Please also provide a brief discussion on how sensitive your results were based on the number of clusters assumed, clustering technique, etc.

Response: We expanded Sections 2.3 and 3.3 by adding more information about the methodology and the results, as follows:

"Section 2.3

To determine the primary sources of NH$_3$ measured at the UNAM station and to assess the dominant atmospheric transport pathways during the events with the largest hourly means of the NH$_3$ columns in

the time series, trajectory cluster analysis (Reizer and Orza, 2018) was applied. Eight-hour back-trajectories were selected to capture the air masses passing over the MCMA. Using the UNAM station as the receptor, back-trajectories were calculated using the Hybrid Single-Particle Lagrangian Integrated Trajectory (HYSPLIT) model (Stein et al., 2015; Draxler et al., 1997) at different altitudes above the UNAM station level (2280 m.a.s.l.). The HYSPLIT model can be run online at the following link https://www.ready.noaa.gov/HYSPLIT.php. The wind data used for the back-trajectories was derived from the NCEP North American Mesoscale (NAM) analysis product at 12 km and 1 hour of spatial and temporal resolution respectively. The cluster analysis is an embedded routine in HYSPLIT and is based on the Ward's agglomerative hierarchical clustering algorithm (Ward, 1963). Finally, the Total Spatial Variance (TSV) method (Draxler et al., 2021), included in HYSPLIT, was used to fit the number of clusters that represent the data.

Section 3.3

A cluster analysis was applied using eight-hour back-trajectories 100 m above UNAM station to identify the main transport pathways for air masses arriving at this station that correspond to the highest average hourly $NH_3$ total columns (Figure 7d). The 100 m cluster was considered the most representative because $NH_3$ is mostly concentrated near the surface. The TSV method was able to represent the primary trajectories at 100 m above UNAM with only three clusters"

We added the reference below.

Ward, J: Hierarchical Grouping to Optimize an Objective Function. Journal of the American Statistical Association; 58, 301, 236-244, doi:10.1080/01621459.1963.10500845, 1963.

**Minor Comments**

Line 71: 'come from'

Response: Done in RC1

Line 74: Suggest restructuring this sentence to − 'The inventory also attributes a meaningful NH3 contribution (%) to a range of different population activities (e.g., X,Y,Z) and feces from domesticated animals'

Response: Done

Line 89: '.. , all classified as ..'

Response: Done in RC1

Line 178: 'The entire period …' – Please restructure this sentence

Response: We have modified this sentence in the revised manuscript as: "The average NH3 total columns for the entire period (1.46± 0.64 x10$^{16}$ molecules/cm$^2$ at UNAM and 1.87 ± 2.40 x10$^{15}$ molecules/cm2 at Altzomoni) are listed and …"

Line 206: '.. conversion to ammonium, as was observed …'

Response: Done

Line 314: 'Comparisons between the seasonal and temporal variability of NH3 over …'

Response: Done

Line 317: 'However, IASI-NH3 shows a …'

Response: Done

Line 318: 'The temporal evolution is represented using ..'

Response: Done

Line 320: 'even in Altzomoni, except for 2013 which was …'

Response: Done

 Line 321: 'averaged'

Response: Done

Line 322: 'molecules/cm2), yields a 62% increase in Mexico City over the course of a decade, in agreement with the trend of …..'

Response: Done and complemented with a comment in RC1 A ", yields a  62 % increase over a decade for Mexico City, in agreement with"

Line 332: Please restructure for clarity

Response: We have modified this sentence in the revised manuscript as:

 "However, the individual back-trajectories that comprise the red cluster (the thin black lines in Figure 7d), indicate that most of the $NH_3$ detected at UNAM comes from a variety of local sources and does not originate exclusively from $NH_3$-enriched air masses transported from the enhancement region to the northeast observed in Figures 7a to 7c."